# Hydrothermal Cobalt Doping of Titanium Dioxide Nanotubes towards Photoanode Activity Enhancement

**DOI:** 10.3390/ma14061507

**Published:** 2021-03-19

**Authors:** Mariusz Wtulich, Mariusz Szkoda, Grzegorz Gajowiec, Maria Gazda, Kacper Jurak, Mirosław Sawczak, Anna Lisowska-Oleksiak

**Affiliations:** 1Department of Chemistry and Technology of Functional Materials, Chemical Faculty, Gdańsk University of Technology, 80-233 Gdańsk, Poland; wturglish@gmail.com (M.W.); mariusz.szkoda1@pg.edu.pl (M.S.); 2Faculty of Mechanical Engineering and Ship Technology, Institute of Machine Technology and Materials, Gdansk University of Technology, 80-233 Gdansk, Poland; grzgajow@pg.edu.pl; 3Department of Solid State Physics, Faculty of Applied Physics and Mathematics, Gdansk University of Technology, 80-233 Gdansk, Poland; maria.gazda@pg.edu.pl; 4Department of Electrochemistry, Corrosion and Materials Engineering, Chemical Faculty, Gdansk University of Technology, 80-233 Gdańsk, Poland; kacper.jurak@pg.edu.pl; 5The Szewalski Institute of Fluid-Flow Machinery, Polish Academy of Sciences, 80-231 Gdańsk, Poland; mireks@imp.gda.pl

**Keywords:** titanium dioxide, hydrothermal modification, photoanode, water splitting, Co-doping

## Abstract

Doping and modification of TiO_2_ nanotubes were carried out using the hydrothermal method. The introduction of small amounts of cobalt (0.1 at %) into the structure of anatase caused an increase in the absorption of light in the visible spectrum, changes in the position of the flat band potential, a decrease in the threshold potential of water oxidation in the dark, and a significant increase in the anode photocurrent. The material was characterized by the SEM, EDX, and XRD methods, Raman spectroscopy, XPS, and UV-Vis reflectance measurements. Electrochemical measurement was used along with a number of electrochemical methods: chronoamperometry, electrochemical impedance spectroscopy, cyclic voltammetry, and linear sweep voltammetry in dark conditions and under solar light illumination. Improved photoelectrocatalytic activity of cobalt-doped TiO_2_ nanotubes is achieved mainly due to its regular nanostructure and real surface area increase, as well as improved visible light absorption for an appropriate dopant concentration.

## 1. Introduction

The greatest challenge for the 21st century is without doubt combatting climate change. Globally, we should reduce fossil-originated energy consumption [1]. In response to the requests of society, it is now vital that new, environmentally friendly energy conversion and storage equipment is available; consequently, there has been a colossal development of research in this topic. The performance of energy conversion and storage electrochemical devices depends on the materials they are made of. The innovative solution in material chemistry is very often based on nanotechnology. Nanostructured materials hold the key to the novel generation of supercapacitors, lithium-ion/sodium-ion batteries, thermoelectric cells, and photoelectrochemical cells (PEC) [2,3,4,5,6,7]. Considering the PEC cell, the heart of the light energy harvesting module is the photoanode [5]. The history of photoanode material development starts half a century ago with the discovery of the photoelectroactivity of the TiO_2_ monocrystal electrode under UV electromagnetic wave illumination [8]. Great effort has been undertaken globally to give us novel semiconductor materials with a narrow bandgap, such as BiVO_4_, Fe_3_O_4_, WO_3_, and their derivatives [9,10,11,12], allowing visible light absorption [7,8,9,10,11,12]. The strategy to increase the photoactivity and electroactivity of TiO_2_ focuses on modifying the position of the electron bands of the material by doping with non-metals [13,14], by metal doping [6,7,15], or the application of surface sensitizers [16]. The importance of morphology was demonstrated by studies of TiO_2_ nanostructures. Early reports on the possibility of growing titanium dioxide nanotubes come from Assefpour-Dezfuly et al., who managed to obtain them from an acidic electrolyte containing fluoride ions and chromic acid brushes a few hundred nanometers thick [17].

Kasuga et al. produced chemically titania nanotubes (TiNTs) of a diameter below 10 nm [18]. Other chemically synthesized nanotubes are mainly tested in lithium batteries [19]. The earliest fully successful attempt of electrochemical synthesis came from Zwilling et al. in 1999, who reported the first self-organized nanotube coating on a Ti substrate obtained by electrochemical Ti anodization in chromic acid electrolytes containing fluoride ions [20]. Since that point, the interest in the method has increased enormously and the technology of producing nanotubes has developed extensively.

Electrochemical methods were improved significantly in the so-called second generation of titania nanotubes, produced in neutral electrolytes [21]. The most popular method nowadays is growing them from viscous mixed solvent electrolytes based on ethylene glycol (EG), with a complexing agent F^¯^ in the form of HF and NH_4_F, often with H_3_PO_4_ [22,23,24]. Application of TiNTs grown on the Ti support includes their use in PEC cells, encapsulation of drugs, air purification, water purification, construction of sensors, and use in electrochemical capacitors [25,26,27,28,29]. TiNTs in photoelectrochemical devices act as UV-Vis light absorbing photoanode material. Although the size of the bandgap of nanotubes is reduced compared to pure bulk anatase TiO_2_ (3.2 eV), a lot of effort has still been exerted by researchers to obtain higher parameters of photoactivity. The absorption edge of measured spectra for TiO_2_ nanotube on Ti substrate is shifted slightly towards the visible range compared to pure bulk anatase TiO_2_ powder. This is due to the fact that the barrier layer present at the TiO_2_ nanotube—Ti substrate interface has rutile crystallites, and the nanotube walls consist of anatase crystallites. The bandgap of the rutile is lower compared to the anatase. The rutile phase at the barrier layer leads to the shifting of the absorption edge to a higher wavelength [30,31]. The presence of Ti^3+^ in the tubular structure of TiO_2_ is also important for the absorption of light. After anodizing, the layers are subjected to a thermal process which causes the formation of oxygen vacancies and reduction of Ti(IV) to Ti(III). This surface-reduced material shows red shift absorption and better electrical conductivity [32]. There are several ways, as with ordinary structures, to increase the photoelectroactivity of nanotubes. Here, we have doping with non-metals [33,34], metal doping [35], and the use of systems with a co-catalyst as reported for Fe_3_O_4_ and CoO_x_ decorated titania nanotubes towards water splitting [36,37].

In this study, we focus on transition metal, namely Co, doping of the TiO_2_ nanotube structure towards water photoanodic oxidation. Previous reports on photocatalytic organic pollutants degradation show that doping the anatase powder with cobalt leads to the enhancement of photoactivity in the process of phenol disposal from water or dye de-coloration [38,39,40]. Very small amounts of cobalt (0.1–3 at %) introduced into the structure contribute to a significant increase in the photocatalytic organic pollutant decomposition efficiency [38].

Since the photocatalytic activity of Co-doped TiO_2_ is documented, one may expect that the material could be in PEC cells as a photoanode. Titania nanotubes, the synthesis of which is easily controlled, were chosen as starting materials. The Co-doping of the TiO_2_ structure is performed using a hydrothermal procedure.

The goal of this work is to demonstrate the influence of the hydrothermal process, leading to a slight introduction of cobalt ions into the nanotubes, in the photoanode process of water oxidation. The novel electrode material is characterized using X-ray powder diffraction (XRD) analysis, Raman spectroscopy, UV-Vis spectroscopy, X-ray photoelectron spectrometry (XPS), scanning electron microscopy (SEM), electrochemical methods: electrochemical impedance spectroscopy (EIS), cyclic voltammetry (CV), linear sweep voltammetry (LSV), and chronoamperometry (CA) in the dark and under solar light illumination. We demonstrated that a small amount of Co (~0.1 at %) inserted hydrothermally into the TiO_2_ nanotubes causes a threefold increase in the photocurrent under solar light illumination. The hydrothermal method turned out to be a very effective and cheap way to obtain a material with high photoactivity.

## 2. Materials and Methods

### 2.1. Apparatus

The microscopic studies were performed using JSM-7800 F (JEOL, Tokyo, Japan) field emission scanning electron microscope on the surface of pure TiO_2_ and with cobalt dopant. The images were analyzed using a beam accelerating voltage at 5 kV. EDX analysis was used for chemical elements detection by a silicon nitride window’s detector (OCTANE ELITE model, EDAX company, Mahwah, NJ, USA).

The crystal structure was determined from the XRD pattern using a diffractometer (Xpert PRO-MPD, Philips, Amsterdam, The Netherlands) with CuKα emission (λ = 1.5406 Å). The crystallites sizes were estimated based on the Scherrer formula and were processed using *Fityk* software [41] via fitting to the Gaussian function. UV-Vis spectra were recorded in the from 200 nm to 800 nm by a Perkin Elmer UV-Vis spectrometer (Lambda 35, Perkin Elmer, Waltham, MA, USA) equipped with the integrating sphere module for reflectance measurements. The Raman analysis was performed using a Raman microscope (InVia, Renishaw, Wotton-under-Edge, UK). Spectra were received using an argon-ion laser emitting at 514 nm. The spectrum of every single point at the sample was recorded as an accumulation of three scans.

XPS measurements of the electrode materials were performed on the Escalab 250Xi device (Thermo Fisher Scientific, Waltham, MA, USA). Al Kα radiation was used. The spectra of elements were analyzed and deconvoluted into components described by an envelope of a Gaussian–Lorentzian sum function with an asymmetry tail supported by the spectrometer commercial software Avantage version 5.973 [42]. The binding energies obtained in the XPS analysis are given relative to the C1s line at 284.6 eV. Traces amounts of cobalt were searched for, therefore the largest possible spot width with a diameter of 650 µm was used.

The electrochemical examinations in the dark of TiO_2_-NTs, Co-TiO_2_-NTs and Co-TiO_2_ samples were carried out with a potentiostat–galvanostat (AutolabPGStat10 with an FRA module, Eco Chemie B.V, Utrecht, The Netherland and Autolab PGSTAT 30, Metrohm Autolab B.V., Utrecht, The Netherlands) in a three-electrode glassy cell with titanium modified foils as the working electrode, Ag/AgCl/0.1 M KCl as the reference electrode, and platinum mesh as the auxiliary electrode. Cyclic voltammetry, linear sweep voltammetry, chronoamperometry and electrochemical impedance spectroscopy were conducted in 0.1 M K_2_SO_4_, purged with argon gas for 30 min before measurements, and kept at 20 ± 0.1 °C.

The flat band (E_fb_) potentials were determined for all materials using impedance measurements by EIS at the frequency range from 20 kHz to 1 Hz for 10 points per decade with 10 mV point–to-point amplitude of the AC signal.

Photoelectrochemical measurements were carried out in a three-electrode glassy cell with a high transmittance quartz window. The studied samples, with an active surface area of 0.8 ± 0.1 cm^2^, remained in the same electrolyte and under conditions as mentioned above. The light source was a 150 W xenon lamp (Osram XBO 150, Quantum Desing, Darmstadt, Germany) equipped with an AM1.5 filter and an automatic shutter that opened every 10 s.

### 2.2. Chemicals

All reagents used in the electrochemical electrolyte preparation were of analytical grade: CoCl_2_·6H_2_O (POCH, Gliwice, Poland), K_2_SO_4_, H_3_PO_4,_ NaNO_3_, ethylene glycol (POCH, Gliwice, Poland), and Ti foils (99.5% metals base, Alfa Aesar, Kandel, Germany). In all experiments, triple distilled water was used.

## 3. Results

### 3.1. Preparation of the Electrode Materials

Nanostructures of titanium dioxide were prepared by electrochemical oxidation of the titanium foil (0.25 mm thick, annealed, area 2 × 2 cm^2^) in the electrolyte containing 0.27 M NH_4_F in a solution containing 5 vol % of water, 3.5 vol % of H_3_PO_4_ (95%) and 91.5 vol % of EG, at 40 V for 2 h [43]. This process of anodization was performed in a glassy cell with the cooling jacket at 20 ± 0.1 °C, using a thermostat (Julabo F-12, Seelbach, Germany). Afterwards samples were cleaned in an ultrasonic bath in a 1:1 solution of acetone and isopropanol. Samples were then rinsed with triple distilled water and dried using the hot stream of air. The anodized and cleaned TiO_2_ films were calcined at 450 °C for 2 h in the air (the heating rate was 2.5 °C/min) and cooled freely. The above-mentioned steps allow to create the titania nanotubes (TiO_2_-NTs) on the surface of titanium [43].

Subsequently, the TiO_2_-NTs sample was immersed in the 50 cm^3^ Teflon chamber with a 30 mL aqueous solution containing: 1.07 g CoCl_2_·6 H_2_O, 2.56 g NaNO_3_ and 75 µL (35–38%) HCl. The container was sealed off inside a steel autoclave and placed in an oven at 100 °C for 24 h. After this hydrothermal treatment, the sample was taken out, rinsed several times in a solution of 1:1 acetone/isopropyl alcohol and in distilled water. This process was performed to obtain Co-doped titania nanotubes (Co-TiO_2_-NTs). The third sample was prepared by anodization of the Ti plate and hydrothermal modification finally followed by thermal treatment in a tube furnace in contact with air at 450 °C for 2 h. This sample is labeled as Co-TiO_2_.

### 3.2. Morphology and Composition of the Samples

#### Scanning Electron Microscope and Energy Dispersive X-Ray Analysis

The imaging of the titanium surface covered with nanotubes is most often carried out using scanning electron microscopy. The use of SEM is effective due to the good conductivity of the semiconducting titanium dioxide [44]. Imaging allows for determining the length of nanotubes, their wall thickness, the diameter of pores, and interpose distances [44]. Today, it is widely used to analyze surfaces by means of scanning electron microscopy. In Figure 1 is an image obtained from the scanning electron microscope showing pure TiO_2_ nanotubes and modified samples. On the left side, marked as A, is an image of pure TiO_2_ nanotubes. The observed neighboring tubes’ edges touch each other. The diameters are marked in the picture and do not exceed 120 nm (117, 87, 102 nm). The tubular shapes are slightly irregular. They probably reflect the arrangement of the rolled plate substrate. This sample was next hydrothermally modified. The image of the modified sample is shown in picture 1B. As can be seen, the material marked as Co-TiO_2_-NTs is different from the unmodified sample 1A. We have separate tubular shapes with slightly greater diameters. Tubes do not touch their edges. Such an electrode material should have a higher real surface area, and therefore higher currents. Now let us look at what happens to the electrode material if we thermally modify (450 °C, 2 h) the sample Co-TiO_2_-NTs (Figure 1B) to obtain sample Co-TiO_2_ (Figure 1C). The change is drastic. The nanotubes cease to exist and are sealed together—we observe an intricate lace, but not nanotubes any more. Figure 1D shows the length of the Co-TiO_2_-NTs nanotubes. Their length does not exceed 215 µm.

The chemical composition of the sample was estimated by the EDX method. Titanium plates covered with oxide layers were analyzed thoroughly. Two samples showed the presence of small amounts of cobalt. EDX examination of the sample after hydrothermal modification Co-TiO_2_-NTs showed the presence of ~0.1 at % of Co. Sample Co-TiO_2_ contained also ~0.1 at % of Co. The distribution of elements is presented in Appendix A. Due to the low content of Co, scanning by EDX does not allow for a reliable measurement of this dopant distribution. As is shown below, both electrodes with traces of cobalt have catalytic and photoelectrocatalytic enhanced activity towards water oxidation in comparison with pure TiO_2_ nanotubes.

### 3.3. Structure

#### 3.3.1. X-Ray Powder Diffraction

XRD analysis was performed to determine the phase composition, crystallinity, and crystallite size of the prepared samples. The results are shown in Figure 2. The diffraction reflections at 35°, 40.1°, and 53° correlate to the (100), (101), and (102) planes of the hexagonal closest packed titanium [45], respectively. All obtained TiO_2_ nanostructures consisted of the pure anatase phase without the brookite or rutile phase. This is confirmed by the observed at 25.3°, 37.9°, 38.4°, 48°, 54°, and 55° corresponding to Miller indices (101), (004), (112), (200), (105), and (211), respectively [46].

Based on the Scherrer formula, defined by the equation:(1)$$D_{hkl}=\frac{K\lambda }{B_{k}cos\theta }$$
where K is a dimensionless shape factor (K = 0.9), λ is the CuKα radiation X-ray wavelength, $B_{k}$ is the line broadening at half the maximum intensity (FWHM) [47], mean crystallite sizes for selected crystallographic planes were determined, and are summarized in Table 1.

There are no reflections that identify cobalt compound or Co metal. According to Peng Jiang et al. [38], no changes in XRD analysis were observed for low atomic cobalt contents. All prepared samples receive nanometer sizes. However, for the Co-doped TiO_2_, a slight increase in crystallite size in the direction perpendicular to the (112) plane was observed. This phenomenon may affect the frequency shifting and broadening of Raman peaks due to the phonon confinement [48]. It cannot be excluded that small quantities of cobalt had built into the nanostructure of TiO_2_.

#### 3.3.2. Raman Spectroscopy

Raman spectra of TiO_2_-NTs, Co-TiO_2_-NTs, and Co-TiO_2_ samples recorded in the spectral range of 100–1200 cm^−1^ are presented in Figure 3. The Raman bands centered at 144, 197, 392, 515, and 633 cm^−1^ corresponds to E_g_ (1), E_g_ (2), B_1g_, A_1g_, and E_g_ (3) modes of the anatase phase, respectively. The origin of the E_g_ bands can be attributed to the symmetric stretching vibration of O–Ti–O, while the B_1g_ band corresponding to the symmetric bending vibration of O–Ti–O and A_1g_ mode can be assigned to the antisymmetric bending vibration of Ti–O–Ti in the anatase phase. The intense sharp band centered near 144 cm^−1^ corresponds to an external symmetric vibration and confirms the formation of long-range order anatase phase [48,49]. The position of this band correlates to the nanocrystalline grain size and the up-shift of its position can be observed with decreasing of the grain size [50,51]. The broadening and up-shifted position of the E_g_ (1) mode observed for Co-TiO_2_-NTs and Co-TiO_2_ samples may be assigned to structural defects resulting from the doping of TiO_2_ with Co ions.

#### 3.3.3. X-ray Photoelectron Spectroscopy

Helpful information about the chemical structure of modified titania is available from XPS studies. The TiO_2_-NTs and Co-TiO_2_-NTs samples were examined and compared. The third sample, Co-TiO_2_, was not photoactive, and thus was not included in XPS tests. The survey spectra from 0 eV to 1350 eV showed the presence of titanium, carbon, oxygen, and nitrogen for both samples, and additionally cobalt for the Co-TiO_2_-NTs sample. The survey spectra made with the 150 eV transition energy showed the difference between the TiO_2_-NTs and Co-TiO_2_-NTs samples (Figure 4). On the latter, the Co2p peaks and the Auger Co LMM peak (713 eV) are noticeable, which proves the presence of cobalt. Both samples contain titanium, carbon, oxygen, and nitrogen. High-resolution scans were performed in the range of Ti2p, C1s, O1s, N1s, and Co2p. A pass energy of 35 eV and 50 repetitions were used. The carbon came from impurities (adventitious carbon) and was used to calibrate the measurement. A small amount of nitrogen may be from the substrates used to produce the samples and is observed in similar measurements [38].

Titanium appears as a doublet of the Ti2p_1/2_ (464.6 eV) and Ti2p_3/2_ (458.9 eV) peaks. These energies testify to the presence of titanium in the +4 oxidation state in titania [52]. The titanium peaks are identical for the TiO_2_-NTs and Co-TiO_2_-NTs samples, not shown.

The amount of cobalt is low compared to the rest of the elements, but it is detectable and analyzable. The measurement clearly shows the Co2p5 and Co2p3 doublet and the so-called shake up peaks just after the main peaks [53]. The energy difference between the doublet peaks for Co is 15.4 eV and is consistent with the XPS databases [54,55].

Measurement for Co2p_3/2_ in cobalt oxide is 780.4 eV, which is in agreement with the binding energy of cobalt. The presence of shake up peaks is confirmed in the literature and proves the existence of Co^2+^ [38,56]. The estimated cobalt atomic percentage from the XPS measurements was ~0.4 at % at 780.4 eV corresponding to Co2p_3/2_, whereas EDX measurements show ~0.1 at %. Higher values of Co content from XPS measurements may suggest that cobalt is accumulated mostly in the near-surface area. It was thus confirmed by XPS that the Co-TiO_2_-NTs sample contained cobalt in the +2 valence state, and although there was relatively little of it, it was detectable.

Our results are consistent with reported data on powder titania modified during the hydrothermal procedure by a small amount of Co [38]. The authors presented a DFT theory-based calculated electronic structure of Co-doped TiO_2_ and indicated that impurity states are introduced into the forbidden band due to the low concentration cobalt doping. They documented that these impurity states are beneficial to the enhancement of visible light absorption and the improvement of photocatalysis efficiency of powder Co-modified titania as experienced with our titania nanotubes.

### 3.4. Reflectance UV-Vis Spectroscopy

In Figure 5A–C, the absorbance, reflectance, and Tauc plot are given for the obtained materials, respectively. As can be seen, obtained electrodes in the ultraviolet region have the highest absorbance, which is typical for TiO_2_ materials. Moreover, the wide absorption band with the maximum at 550 nm could originate from the presence of sub-bandgap states caused by the structure of the titania nanotube that can trap radiation inside the tube [57]. In the case of the Co-TiO_2_ sample, the absorbance in the visible range (450–700 nm) is characterized by a lower intensity in comparison with the bare TiO_2_ and Co-TiO_2_-NTs. This means that absorption in this range of visible light, as clearly seen, is more intense for the samples with the nanotubular structure. The TiO_2_ material characterized by a non-tubular structure does not exhibit this feature. In the case of Co-TiO_2_-NTs, the redshift of the absorption edge is observed, which is advantageous for the materials expected to be photoactive under visible light illumination. The dopant presence is responsible for changes, very likely due to impurities formed in between the forbidden energy bandgap [38]. On the other hand, physicochemically bond water is also supposed to cause red shift [58]. Taking into account the fact that Co-TiO_2_-NTs were not subjected to a thermal process after hydrothermal treatment, the second argument seems to be very likely prevailing.

Based on the Tauc plot, the bandgap energy values were determined and are 2.85 eV, 2.92 eV, and 2.99 eV for Co-TiO_2_, Co-TiO_2_-NTs, and pure TiO_2_, respectively. Therefore, both modified materials were characterized by a slightly narrower bandgap energy in comparison with pure titania nanotubes. Although Co-TiO_2_ has the smallest energy bandgap, it has the worst photoelectrochemical activity compared to the other materials. The reason for this is very likely the lack of a tubular structure and low absorption in the visible light range as recorded on UV-Vis spectra (see Figure 5A).

### 3.5. Electrochemical and Photoelectrochemical Properties

The electrode materials TiO_2_-NTs, Co-TiO_2_-NTs, and Co-TiO_2_ were subjected to electrochemical studies in the dark in the potential range from −0.7 V to 1.3 V versus Ag/AgCl/0.1 M KCl in 0.1 M K_2_SO_4_. Polarization towards high anode potentials shows whether there are differences in the threshold potentials for oxidation of the water molecule. As you can see, the course of the linear sweep voltammetry curves differs (Figure 6). The threshold potentials E_th_ have been determined and their values increase for the electrodes in the order Co-TiO_2_-NTs < Co-TiO_2_ < TiO_2_-NTs with 100 mV lowering of the E_th_ value for Co-doped nanotubes.

As can be seen, the introduction of cobalt ions into the structure of titanium dioxide, as well as morphological changes after modification by the hydrothermal method, definitely affect the electrochemical properties of the new systems. The environment of cobalt ions in the structure is to some extent similar to that of ordinary cobalt oxide compounds. Such compounds belong to a group known for their catalytic properties in relation to the oxidation of water molecules [59]. Undoubtedly, both electrodes containing a small amount of cobalt catalyze the oxidation of water to molecular oxygen. The decrease in the water oxidation threshold potential *E_t_*_h_ for the modified electrode indicates the effectiveness of the hydrothermal process.

TiO_2_ nanotubes are known for their cathode potential activity. Their typical course of CV curves obtained for pure and modified titania are shown in Figure 7A–C. The CV curves obtained for different sweep rates of polarization allow the diagnosis of the reaction mechanism. Linear regression for changes in current with the square root of the polarization sweep rate indicates diffusive process control. When current intensity changes linearly with the polarization rate *v*, one deals with kinetic process control, and the charge transfer reaction is surface limited (see Figure 7D–F). However, the most important outcome from these simple polarization measurements is the possibility of comparing the effective current densities. These values are calculated in relation to the geometric surface area. At the same chosen potential (−0.5 V), the highest cathode currents were recorded for the Co-TiO_2_-NTs electrode, and the lowest were recorded for the Co-TiO_2_ electrode. Considering the microscopic morphology studies showing that each nanotube has a free surface along its entire length, and direct contact between the tubes is almost absent for the Co-TiO_2_-NTs material, this condition compared to the pure TiO_2_-NTs and calcined Co-TiO_2_ represents an increased real surface area. Hence, the recorded current for Co-TiO_2_-NTs is the highest. The lowest current densities are recorded for electrodes after final calcination following hydrothermal modification. This material is devoid of a nanotubular structure—it is the result of calcination. The Co-TiO_2_ sample has an undeveloped specific surface area, and thus the current density related to the geometric surface area is lower. Thus, the research on polarization in terms of cathode potentials confirmed the typical activity of TiO_2_ nanotubes, as was reported in Ref. [60]. Moreover, it allowed us to assess the state of surface development for a hydrothermally modified electrode. The modification was considered to increase the real surface area due to hydrothermal treatment.

Titanium dioxide belongs to the n-type semiconductors. Any changes in structure—doping, the presence of surface modifying substances—will influence the position of the measured value of the flat band potential, the most important value characteristic for the semiconductor electrode [61].

The flat band potential E_fb_ was determined for three electrodes: TiO_2_-NTs, Co-TiO_2_-NTs, and Co‑TiO_2_ materials. Electrochemical impedance spectroscopy was used to measure the impedance of the polarized samples under stationary conditions at a frequency range from 20 kHz to 1 Hz, and AC signal amplitude 10mV. Examples are gathered in Figure 8A–C. The impedance functions presented in the figures are of a typical capacitive character, the course of the function is an almost vertical line, slightly rotated, in line with the presence of *CPE* (constant phase element), which is neglected here. The capacitance *C* values were calculated for frequencies: 1 Hz, 100 Hz, and 1000 Hz using the simple relation: −*Z*” = (*j*ωC)^−1^, where *C* is the capacitance, Z” is the imaginary part of the impedance function, ω is the angular frequency, *j* =$\;\sqrt{-1}$.

The E_fb_ values were determined in accordance with the Mott–Schottky function, using *C* values extracted from the impedance measurements. The results are presented in Figure 8D–F. Table 2 gathers evaluated flat band potential for different materials. As can be seen, doping nanotubes with cobalt by hydrothermal treatment changes the effective flat band potential significantly—the difference is about 200 mV towards anodic values. This change will be important for the course of the electrode reactions and the availability of the electrons in the conduction band.

According to the aim of our study, the photoelectroactivity of the material in the field of photoanode operation has been investigated. As mentioned and proved above, oxidation of water in dark conditions occurs more easily for the doped electrode, and the E_th_ potential is lowered for the Co-TiO_2_-NTs material. Polarization under solar light illumination was carried out to show how the electrode material behaves as a photoanode.

In Figure 9, the photocurrent generation recorded at a constant potential of 0.5 V vs. Ag/AgCl/0.1 M KCl is given for the pure TiO_2_ nanotubes and the Co-modified electrodes. Of all the samples, the material labeled as Co-TiO_2_ generates the lowest photocurrent. Such weak photoactivity for modified titania prepared by the hydrothermal method and finished by extra is caused by the non-tubular morphology characterized by lower electron lifetimes than the ordered layer, and does not provide good pathways for electron transport [62,63]. Additionally, Co-TiO_2_ has the lowest absorbance intensity compared to other samples in the VIS region, which is considered a very important factor, influencing photoconversion efficiency. The highest photocurrent was observed for Co-TiO_2_-NTs reaching about 40 μA∙cm^−2^ stabilized at 32 μA∙cm^−2^ under sunlight illumination.

The difference between the current registered for a sample in the dark and under its illumination (Δ*j*) are listed in Table 3 together with the ratio between the current registered for a sample in the dark and under its illumination (*j_l_*/*j_d_*) and the ratio between photocurrent and light source optical power (photoresponsivity).

LSV measurements were presented also to compare photoactivity of the obtained electrodes. The obtained results are presented in Figure 10. In the early stages, the photocurrent densities in all examined samples increased with increasing potential applied to the photoanode and then stabilized at a potential of around 0 V vs. Ag/AgCl/0.1 M KCl. The highest photocurrent generation value was obtained for Co-TiO_2_-NTs. At a potential equal to +0.2 V, the value of the current density is more than two times higher than the current registered for pure TiO_2_ nanotubes and sintered Co-TiO_2_. Therefore, the Co species on and in the tubular matrix play an important role in the enhanced photocurrent generation (therefore in the case of non-tubular morphology we do not see an improvement in the photocurrent). Thus, the increase in the charge generation is influenced by the presence of Co and this can be caused by the enhanced light absorption in the visible range, as well as the decrease in the recombination rate and the charge transfer resistance.

The low energy bandgap and photocurrent enhancement are the most important parameters to describe novel photoactive materials. Table 4 shows a comparison of selected doped TiO_2_ materials obtained with different synthesis methods. Due to the various modifications of nanostructure, there are large discrepancies in photocurrent values. However, as the authors mentioned [64,65,66], introducing low amounts of chemical elements into the anatase structure promotes increasing the photoresponse. Too high doping level of TiO_2_ causes a decrease in photocurrent density. The photocurrent for Co-TiO_2_-NTs obtained in this work is similar to the value in article [67]. The energies bandgap for doped titania nanotubes are in the range from 2.82 eV to 3.09 eV [64,68,69]. Thin films of TiO_2_ with Ag or Cu dopant achieve lower Eg values, but they also have worse photoelectroactivity.

As shown above, the cobalt doping of TiO_2_ nanotubes has an effect on both the reduction in the overpotential of water oxidation under dark conditions and the photooxidation process. This increase in photoactivity is the result of amplified absorption of visible light, a narrowing of the bandgap, and effective increase in the real surface area, all as a result of hydrothermal treatment in CoCl_2_ electrolyte.

## 4. Conclusions

A small amount of cobalt 0.1 at % has been successfully doped into TiO_2_ nanotubes using the hydrothermal method. Comprehensive characterization (XPS, XRD, EDX, Raman) confirms the introduction of Co^2+^ into the anatase crystal structure. Hydrothermal treatment of TiO_2_-NTs in CoCl_2_ electrolyte leads to an increase in the real surface area of the tubular structure. Titania is modified considering its morphology and electronic structure. Development of the real surface area and presence of Co-dopant are responsible for the significant rise in UV-Vis light absorption. Novel electrode material exhibits increased electroactivity towards water oxidation in the dark as an anodic threshold potential E_th_ is 100 mV lower in comparison with E_th_ for pure nanotubes. Moreover, the photoelectrocatalytic activity towards water oxidation is significantly enhanced, as a photocurrent of the modified sample Co-TiO_2_-NTs is almost threefold higher in comparison to the photocurrent of pure nanotubes. The photoelectrocatalytic and electrocatalytic activity of cobalt-doped TiO_2_ nanotubes is attributed to their real surface increase and improved visible light absorption and, above all, their appropriate doped concentration. The presence of Co is necessary to reduce water oxidation potential in the dark and contributes to the growth of photocurrent.

Although we are aware that EDX elemental analysis of the samples combing both Ti support and the thin film only shows estimated values of Co-dopant, it is evident that hydrothermal low-temperature procedure leads to the formation of better photoanodic material. The further thermal treatment causes destruction of the tubular structure and leads to a hefty reduction in photocurrents while maintaining the catalytic properties for OER reactions in the dark. Our results are consistent with previously published data on photocatalytic activity of Co-doped titania [38,72].

## Figures and Tables

**Figure 1 materials-14-01507-f001:**
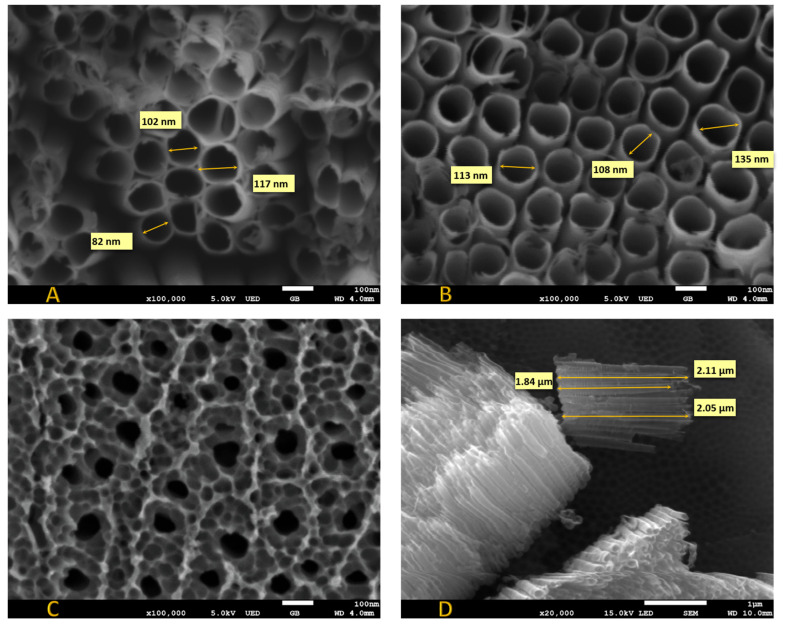
SEM images: (**A**) pure TiO_2_ nanotubes TiO_2_-NTs; (**B**) sample hydrothermally modified Co-TiO_2_-NTs; (**C**) sample Co-TiO_2_ after final thermal treatment (450 °C, 2 h); and (**D**) length of nanotubes for Co-TiO_2_-NTs.

**Figure 2 materials-14-01507-f002:**
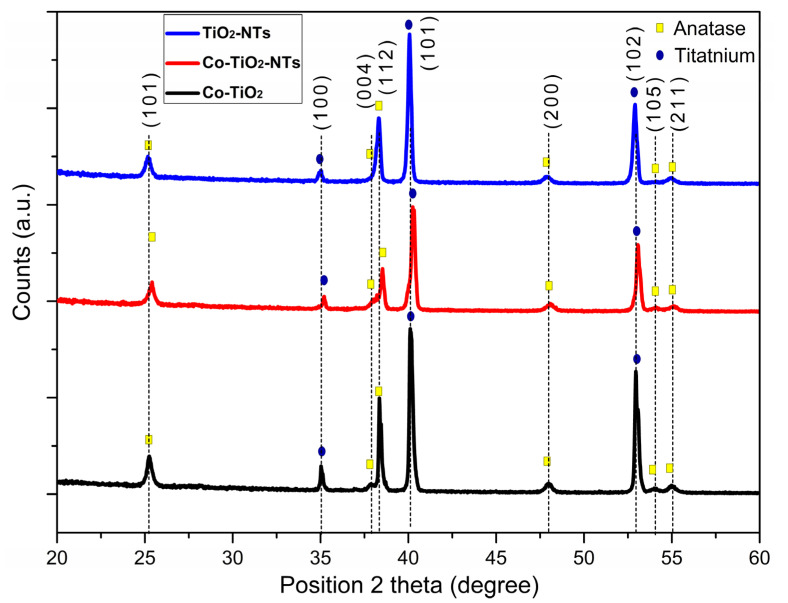
Comparison of XRD patterns of titanium samples covered with TiO_2_ nanotubes pure and with cobalt doping.

**Figure 3 materials-14-01507-f003:**
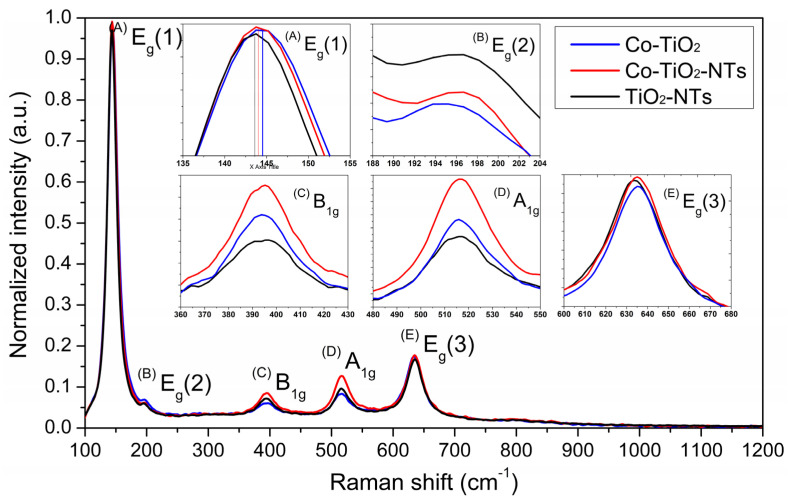
Raman spectra of pure titania nanotubes and Co-doped TiO_2_ samples. (**A–E**) typical modes of the anatase phase.

**Figure 4 materials-14-01507-f004:**
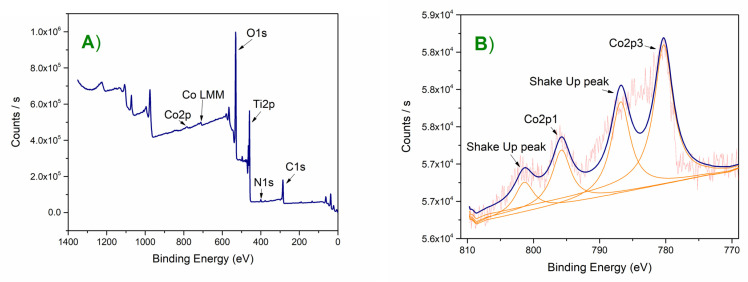
(**A**) Survey spectrum. It is almost identical for both samples. The Co-TiO_2_-NTs sample additionally reveals cobalt in the form of Co2p and Co LMM peaks; (**B**) detailed cobalt spectrum for Co-TiO_2_-NTs.

**Figure 5 materials-14-01507-f005:**
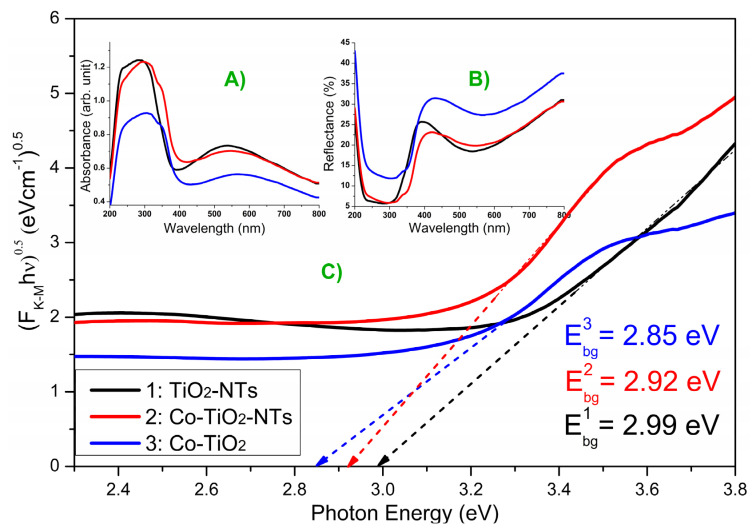
(**A**) Absorbance, (**B**) reflectance spectra, and (**C**) Tauc plot for samples TiO_2_-NTs, Co-TiO_2_-NTs, Co-TiO_2_.

**Figure 6 materials-14-01507-f006:**
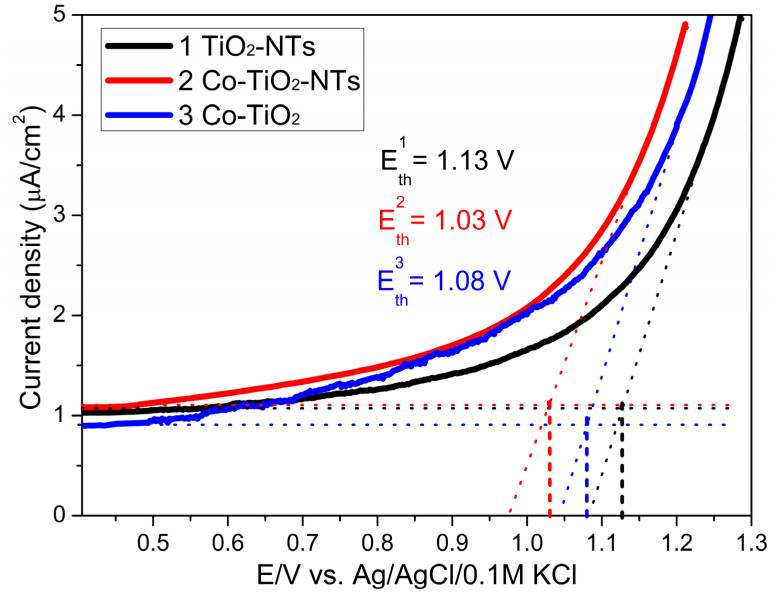
Linear sweep voltammetry (LSV) curves obtained for samples TiO_2_-NTs, Co-TiO_2_-NTs, and Co-TiO_2_, used for the evaluation of the water oxidation threshold potential E_th_, sweep rate 50 mV·s^−1^.

**Figure 7 materials-14-01507-f007:**
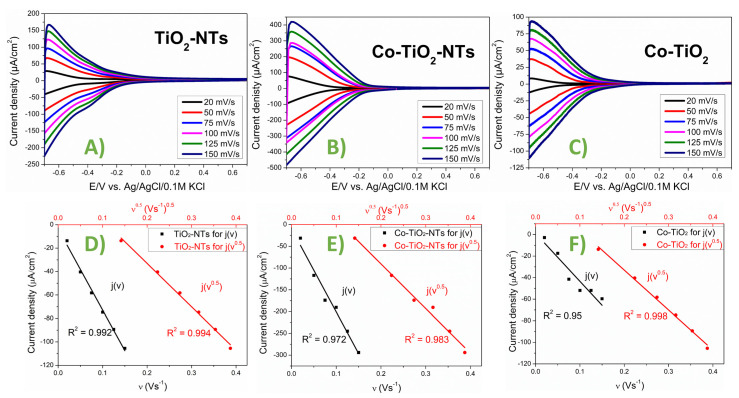
Cyclic voltammetry (CV) curves obtained at varied sweep rates and analysis: (**A**,**D**) TiO_2_-NTs; (**B**,**E**) CoTiO_2_-NTs; (**C**,**F**) Co-TiO_2_.

**Figure 8 materials-14-01507-f008:**
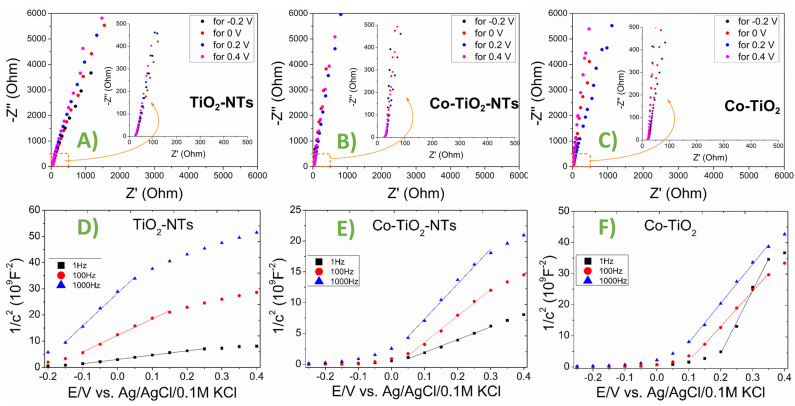
Complex plane impedance plots obtained for: (**A**) TiO_2_-NTs, (**B**) CoTiO_2_ and (**C**) Co-TiO_2_, frequency range 20 kHz to 1 Hz. Mott–Schottky plots for: (**D**) TiO_2_-NTs, (**E**) CoTiO_2_, (**F**) Co-TiO_2_.

**Figure 9 materials-14-01507-f009:**
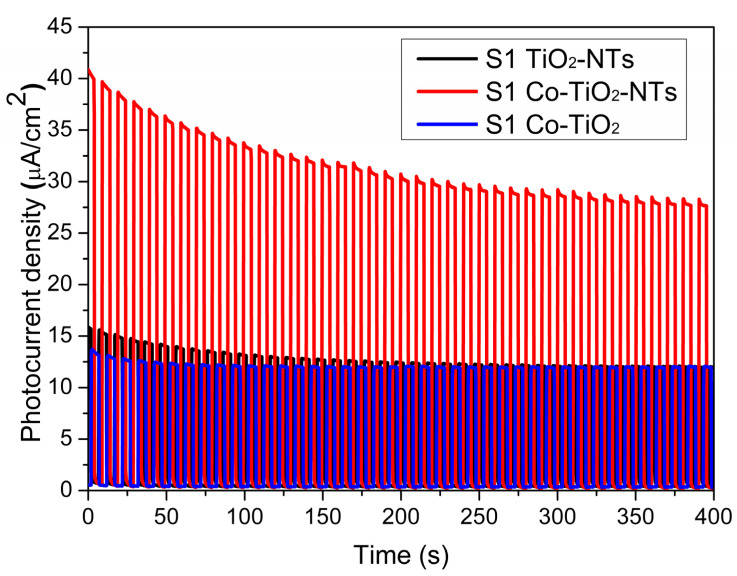
The chronoamperometry curve, showing the photocurrent density, registered for modified and pure titania electrodes at E = + 0.5V. Source of light: 150 W xenon lamp equipped with an AM1.5 filter, light intensity: 100 mW·cm^2^.

**Figure 10 materials-14-01507-f010:**
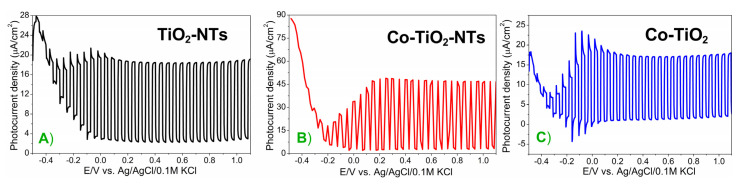
The linear sweep voltammetry curves obtained at a sweep rate 50 mV·s^−1^ for samples (**A**) TiO_2_‑NTs (black line), (**B**) Co-TiO_2_-NTs (red line), (**C**) Co-TiO_2_ (blue line) in the 0.1 M K_2_SO_4_ aqueous electrolyte. Source of light: 150 W xenon lamp equipped with an AM1.5 filter, light intensity: 100 mW·cm^2^.

**Table 1 materials-14-01507-t001:** Mean crystallite sizes calculated from the Scherrer’s equation based on achieved XRD patterns for TiO_2_-NTs, Co-TiO_2_-NTs and Co-TiO_2_.

Sample	D_101_ (nm)	D_112_ (nm)	D_200_ (nm)	D_211_ (nm)
TiO_2_-NTs	18.2	28.2	16.5	25.5
Co-TiO_2_-NTs	15.3	41.1	14.6	17.5
Co-TiO_2_	19.7	34.6	15.0	13.3

**Table 2 materials-14-01507-t002:** Flat band potential values for selected frequency for pure and Co-doped TiO_2_ samples.

Sample	E_fb_ (V) for 1 Hz	E_fb_ (V) for 100 Hz	E_fb_ (V) for 1000 Hz
TiO_2_-NTs	−0.21 ± 0.02	−0.22 ± 0.02	−0.25 ± 0.02
Co-TiO_2_-NTs	−0.02 ± 0.02	0.00 ± 0.02	0.00 ± 0.02
Co-TiO_2_	0.15 ± 0.02	0.06 ± 0.02	0.02 ± 0.02

**Table 3 materials-14-01507-t003:** Determined photoresponsivity values from Figure 9, where: Δ*j* is the difference between the current registered for a sample in the dark and under its illumination, *j_l_*/*j*_d_ is the ratio between the current registered for a sample in the dark and under its illumination, and Δ*j/P* (photoresponsivity) is the ratio between photocurrent and light source optical power.

Sample	Δ*j* (μA·cm^−2^)	*j_l_*/*j*_d_	Δ*j/P* (μA∙W^−1^)
TiO_2_-NTs	12.51	33.08	12.5
Co-TiO_2_-NTs	32.71	137.29	32.7
Co-TiO_2_	11.68	33.44	11.7

**Table 4 materials-14-01507-t004:** Comparison of energy bandgap and photocurrent density for selected metal-doped TiO_2_ (*j*_doped_—photocurrent density for doped TiO_2_, *j_TiO_**_2_*—photocurrent density for pure TiO_2_).

Electrode Material	Luminous Intensity (mW·cm^2^)	Energy Bandgap, E_g_ (eV)	Photocurrent Density (μA·cm ^2^)/E * (V)	Enhancement Factor (*j*_doped_/*j*_TiO__2_)	Ref.
TiO_2_-NTs	100	2.99	12.9 at 0.5 V	1	This work
Co-TiO_2_		2.85	12.0 at 0.5 V	0.9	
Co-TiO_2_-NTs		2.92	33.3 at 0.5 V	2.6	
Co-TiO_2_-NTs	100	3.09	95.0 at 0.5 V	1.5	[68]
Co-TiO_2_-NTs	100	no data	40.0 at 0.4 V	3.0	[67]
Cr-TiO_2_-NTs	100	2.82	360.0 at 1.0 V	9.2	[64]
B-TiO_2_-NTs	100	2.91	311.0 at 0.5 V	7.4	[69]
V-TiO_2_-NTs	16	no data	5.8 at 0.5 V	4.8	[70]
Ag-TiO_2_ film	4.4	2.5	1.2 at 0.2 V	3.5	[71]
Fe-TiO_2_ nanorods	100	3.12	550.0 at 0 V	5.5	[65]
Cu-TiO_2_ film	44.42	2.82	18.2 at 0.4 V	1.3	[66]

*: Electrode potential E vs. Ag/AgCl.

## Data Availability

Data is contained within the article or supplementary material.

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
