# Peer review of "Hydrothermal Cobalt Doping of Titanium Dioxide Nanotubes towards Photoanode Activity Enhancement"

_materials, 2021, doi:10.3390/ma14061507_

Round 1
Reviewer 1 Report
In this report, anatase TiO2 nanotubes was successfully growth on titanium foil by anodization and then treated hydrothermally in presence of cobalt chloride in order to obtain the Co-doped TiO2 nanotubes. The material possesses a narrower band gap of 2.85 eV, phase purity, and morphological homogeneity. Regarding practical applications, the product exhibits enhanced electroactivity towards water oxidation in dark and increased photoelectrocatalytic activity in water. Both the originality and novelty are presented in the paper. On the other hand, some questions must be clarified before possible acceptance for publication, as specified below.
- The cobalt concentration in sample reported in this paper (0.1 at.%) was only from EDX method. I suggest to discuss this moment in more detail using the values from synthesis (planned concentration) and XPS measurements.
- EDX mapping may help to confirm the uniform distribution of Co in nanotubes. If possible, I suggest to provide such experiments.
- Significance of these Co-doped TiO2 nanotubes should be clearly demonstrated. The authors should summarize the results (may be in tabular form) and compare the main tested parameters with that from literature for doped TO2.
- Some recent relevant papers on doped titanium dioxide nanostructures for energy storage and conversion are suggested to cite in the Introduction:
10.1016/j.jmst.2020.02.068
10.1016/j.chemphys.2020.110864
As I believe, the revised manuscript that addresses these concerns may be acceptable for publication in the Materials.
Author Response
We would like to thank the Reviewers for their efforts to improve our work. All comments were taken into account in the preparation of the revised version and we are grateful for a thorough reviews.
Reviewer 1
In this report, anatase TiO2 nanotubes was successfully growth on titanium foil by anodization and then treated hydrothermally in presence of cobalt chloride in order to obtain the Co-doped TiO2 nanotubes. The material possesses a narrower band gap of 2.85 eV, phase purity, and morphological homogeneity. Regarding practical applications, the product exhibits enhanced electro-activity towards water oxidation in dark and increased photoelectrocatalytic activity in water. Both the originality and novelty are presented in the paper. On the other hand, some questions must be clarified before possible acceptance for publication, as specified below.
- The cobalt concentration in sample reported in this paper (0.1 at.%) was only from EDX method. I suggest to discuss this moment in more detail using the values from synthesis (planned concentration) and XPS measurements.
A1. Thank you for pointing out the possibility to combine EDX and XPS results in respect to Co content in the film. The cobalt atomic percentage from the XPS is equal to 0.4 at%. In revised version we added lines:
Line 270- 273 :
The estimated cobalt atomic percentage from the XPS measurements was ~0.4 at% at 780.4 eV corresponding to Co2p3/2, whereas EDX measurements show ~0.1 at%. Higher values of Co content from XPS measurements may suggest that cobalt is accumulated mostly in the near-surface area
- EDX mapping may help to confirm the uniform distribution of Co in nanotubes. If possible, I suggest to provide such experiments.
A2. Thank you for indicating the method of confirming the dopant distribution in the material. We have carried out tests, but too low a concentration does not allow identification of the distribution for Co. The tests were carried out for 0.1 at% and 0.3 at% concentrations, in both cases the analysis covered an area of 10x10 [µm]. The obtained image does not allow showing the distribution of Co in the sample. Signals recorded for Co (S1) are very likely artefacts from the instrument. The concentration should be above 5 at% to give the unambiguous image. We did not achieve such content and from our photoelectrochemical measurements low concentration ~ 0.1 at% is the best for photocatalytic application.
Line 198-200
The distribution of elements is presented in Supplementary Materials (S1). Due to the low content of Cobalt, EDX scanning does not allow for a reliable measurement of this dopant distribution.
Line 476-477
We attached supplementary results S1.
- Significance of these Co-doped TiO2 nanotubes should be clearly demonstrated. The authors should summarize the results (may be in tabular form) and compare the main tested parameters with that from literature for doped TiO2.
A3. Thank you for your very valuable suggestion. The text below has been added to the manuscript:
Line:431-444
The low energy band-gap and photocurrent enhancement are the most important parameters to describe novel photoactive materials. Table 4 shows a comparison of selected doped TiO2 materials obtained with different synthesis methods. Due to the various modification of nanostructure, there are large discrepancies in photocurrent values. However, as the authors mentioned [66,70-71], introducing low amounts of chemical elements into the anatase structure promotes increasing the photoresponse. The too high a doping level of TiO2 causes a decrease of photocurrent density. The photocurrent for Co-TiO2-NTs obtained in this work is similar to the value in the article [65]. The energies band-gap for doped titania nanotubes are in the range from 2.82 to 3.09 eV [64,66-67]. Thin films of TiO2 with Ag or Cu dopant achieve lower Eg values, but they also have worse photoactivity.
Table 4. Comparison of energy band-gap and photocurrent density for selected metal –doped TiO2 (jdoped – photocurrent density for doped TiO2, jTiO2 – photocurrent density for pure TiO2).
|
Electrode material |
Luminous intensity [mW· cm-2] |
Energy band-gap, Eg [eV] |
Photocurrent density [μA·cm-2]/E*,[V] |
Enhancement Factor [jdoped/jTiO2] |
Ref. |
|
TiO2-NTs |
|
2.99 |
12.9 at 0.5 |
1 |
This work |
|
Co-TiO2 |
100 |
2.85 |
12.0 at 0.5 |
0.9 |
|
|
Co-TiO2-NTs |
|
2.92 |
33.3 at 0.5 V |
2.6 |
|
|
Co-TiO2-NTs |
100 |
3.09 |
95.0 at 0.5 V |
1.5 |
[64 |
|
Co-TiO2-NTs |
100 |
no data |
40.0 at 0.4 V |
3.0 |
[65] |
|
Cr-TiO2-NTs |
100 |
2.82 |
360.0 at 1.0 V |
9.2 |
[66] |
|
B-TiO2-NTs |
100 |
2.91 |
311.0 at 0.5 V |
7.4 |
[67] |
|
V-TiO2-NTs |
16 |
no data |
5.8 at 0.5 V |
4.8 |
[68] |
|
Ag-TiO2 film |
4.4 |
2.5 |
1.2 at 0.2 V |
3.5 |
[69] |
|
Fe-TiO2 nanorods |
100 |
3.12 |
550.0 at 0 V |
5.5 |
[70] |
|
Cu-TiO2 film |
44.42 |
2.82 |
18.2 at 0.4 V |
1.3 |
[71] |
*/Electrode potential E vs. Ag/AgCl
As shown above, the cobalt doping of TiO2 nanotubes has an effect on both the reduction of the overpotential of water oxidation under dark conditions and the photooxidation process. This increase in photoactivity is the result of amplified absorption of visible light, a narrowing of the band gap, and effective increase in the real surface area, all as a result of hydrothermal treatment in CoCl2 electrolyte.
- Some recent relevant papers on doped titanium dioxide nanostructures for energy storage and conversion are suggested to cite in the Introduction:
10.1016/j.jmst.2020.02.068
10.1016/j.chemphys.2020.110864
A4. Thank very much for pointing other fields of possible applications for doped TiO2. References to the articles on Titania nanotubes modification and its application in Li-ion and Na –ion batteries are introduced into Introduction. We will keep this literature for our work on Li-ion batteries as well.
Thus references are given as [6, 7] in line: 39.

Reviewer 2 Report
This paper reports the titanium dioxide nanotubes (TNTs) doped with cobalt (Co) using a hydrothermal method and their structural, optical, electrochemical, and photoanode activity were systematically investigated using SEM, EDX, XRD, Raman spectroscopy, XPS, UV-Vis spectrometry, chronoamperometry, electrochemical impedance spectroscopy, cyclic voltammetry, and linear sweep voltammetry. This paper is relatively well organized and written. But, there is missing information. After revision, therefore, I recommend this paper for publication to Materials. My suggestions are the following:
1. Line 64: What is the term ‘TNTs’?
2. Lines 67-68: The authors need to further provide the detailed reason why the bandgap of TiO2 nanotubes is reduced to be ~2.8 eV, compared to pure bulk anatase TiO2 (3.2eV).
3. Abbreviations, such as XRD, SEM, XPS, CV, LSV, CA, etc., are used repeatedly throughout the manuscript.
4. In section 3.2.1, ‘scanning electron microscope’ should be expressed to ‘SEM’.
5. In section 3.4, the authors need to discuss the reason why the absorption edge of Co-TiO2-NTs is red-shifted, compared to the TiO2 NTs.
6. In Figure 9, further detailed measurement information of photocurrent density should be provided, including optical power density, wavelength.
7. For the photoanode activity of the sample proposed in this work, the photoresponsivity and light-on and -off ratio can be useful. Two data can be obtained from Figure 9.
8. Overall, there are mistakes on grammar and spelling in the manuscript. Fine/minor spell check in English language and style should be strongly required.
1) Line 49: “of to growing” → “of growing”.
2) Line 101: “(EDS)” → “(EDX)”.
3) Line 129: “0,1” → “0.1”.
4) Line 136: “150 A” → “150 W”.
5) Line 148: “V voltage” → “V”.
6) Line 210: “0,9” → “0.9”.
7) Line 227: “100÷1000” → “100-1200”.
Author Response
Reviewer 2
This paper reports the titanium dioxide nanotubes (TiNTs) doped with cobalt (Co) using a hydrothermal method and their structural, optical, electrochemical, and photoanode activity were systematically investigated using SEM, EDX, XRD, Raman spectroscopy, XPS, UV-Vis spectrometry, chronoamperometry, electrochemical impedance spectroscopy, cyclic voltammetry, and linear sweep voltammetry. This paper is relatively well organized and written. But, there is missing information. After revision, therefore, I recommend this paper for publication to Materials. My suggestions are the following:
- Line 64: What is the term ‘TNTs’?
A1. Thank you, the change has been introduced to the manuscript. :
Line 52
titania nanotubes (TiNTs)
- Lines 67-68: The authors need to further provide the detailed reason why the bandgap of TiO2 nanotubes is reduced to be ~2.8 eV, compared to pure bulk anatase TiO2 (3.2eV).
A2. Thank you very much for pointing out inaccuracies in the description. At the same time, we apologize for the error (line VV is : 2.8 eV, should be 2.98 / 3.0 eV) for pure non-doped titania nanotubes. We have added information about the change of the band gap energy in the Introduction. Moreover, the Eg values are included in Table 4 (revised version) for choices metal-doped TiNTs.
The text added to the Introduction part:
Lines 68-77
The absorption edge of measured spectra for TiO2 nanotube on Ti substrate is shifted slightly towards visible range compared to pure bulk anatase TiO2 powder. This is due to the fact that the barrier layer present at TiO2 nanotube/Ti substrate interface has rutile crystallites and the nanotube walls consist of anatase crystallites. The bandgap of the rutile is lower compared to the anatase. The rutile phase at the barrier layer leads to the shifting of the absorption edge to higher wavelength [30-31]. The presence of Ti3+ in the tubular structure of TiO2 is also important for the absorption of light. After anodizing, the layers are subjected to a thermal process which causes the formation of oxygen vacancies and reduction of Ti(IV) to Ti(III). This surface-reduced material shows red shift absorption and and better electrical conductivity [32].
- Abbreviations, such as XRD, SEM, XPS, CV, LSV, CA, etc., are used repeatedly throughout the manuscript.
A3. The abbreviations have been standardized in the manuscript.
- In section 3.2.1, ‘scanning electron microscope’ should be expressed to ‘SEM’.
A4. The abbreviation has been standardized in the manuscript.
- In section 3.4, the authors need to discuss the reason why the absorption edge of Co-TiO2-NTs is red-shifted, compared to the TiO2
A5 Thank you for pointing out this important issue. Partially this issue is discussed in relation to DFT calculations [38] in section 3.3. We extended this by adding lines in section 3.4 as should be in the first place.
Line 297-303:
In the case of Co-TiO2-NTs, the redshift of the absorption edge is observed, which is advantageous for the materials expected to be photoactive under visible light illumination. The dopand presence is responsible for changes very likely due to impurities states formed in between forbidden energy band gap [ 38]. On the other hand physicochemically bond water is also supposed to cause red-shift [58]. Taking into account the fact that Co-TiO2NTs were not subjected to a thermal process after hydrothermal treatment, the second argument seems to be very likely and prevailing.
- In Figure 9, further detailed measurement information of photocurrent density should be provided, including optical power density, wavelength.
A6. According to Referee’s suggestion, we added the necessary information under the Fig. 9. and in Materials and Methods
Line 142-143.
The light source was a 150 W xenon lamp (Osram XBO 150) equipped with an AM1.5 filter and an automatic shutter that opened every 10 seconds.
- For the photoanode activity of the sample proposed in this work, the photoresponsivity and light-on and -off ratio can be useful. Two data can be obtained from Figure 9.
A7. According to Referee’s suggestion, we added additional data as shown in Table 3.
The text added to the manuscript:
Lines 406-414
The difference between the current registered for a sample in the dark and under its illumination (Δj) are listed in Table 3 together with the ratio between the current registered for a sample in the dark and under its illumination (jl/jd) and the ratio between photocurrent and light source optical power (photoresponsivity).
Table 3. Determined photoresponsivity values from Figure 9, where: * Δj – the difference between the current registered for a sample in the dark and under its illumination, ** jl/jd - the ratio between the current registered for a sample in the dark and under its illumination, *** Δj/P - the ratio between photocurrent and light source optical power .
|
Sample |
Δj* [μA·cm-2] |
jl/jd** |
Photoresponsivity*** [μA∙W-1] |
|
TiO2-NTs |
12.51 |
33.08 |
12.5 |
|
Co-TiO2-NTs |
32.71 |
137.29 |
32.7 |
|
Co-TiO2 |
11.68 |
33.44 |
11.7 |
- Overall, there are mistakes on grammar and spelling in the manuscript. Fine/minor spell check in English language and style should be strongly required.
1) Line 49: “of to growing” → “of growing”.
corrected
2) Line 101: “(EDS)” → “(EDX)”.
corrected
3) Line 129: “0,1” → “0.1”.
corrected
4) Line 136: “150 A” → “150 W”.
corrected
5) Line 148: “V voltage” → “V”.
6) Line 210: “0,9” → “0.9”.
7) Line 227: “100÷1000” → “100-1200”.
A8. Thank you very much for your help, all indicated mistakes are removed an English spelling and grammar correction are made.

Round 2
Reviewer 1 Report
I think that the manuscript has been improved satisfactory and now it is acceptable for publication.